# Momentum transfer from the DART mission kinetic impact on asteroid Dimorphos

Andrew F. Cheng[1✉], Harrison F. Agrusa[2], Brent W. Barbee[3], Alex J. Meyer[4], Tony L. Farnham[2], Sabina D. Raducan[5], Derek C. Richardson[2], Elisabetta Dotto[6], Angelo Zinzi[7,8], Vincenzo Della Corte[9], Thomas S. Statler[10], Steven Chesley[11], Shantanu P. Naidu[11], Masatoshi Hirabayashi[12], Jian-Yang Li[13], Siegfried Eggl[14], Olivier S. Barnouin[1], Nancy L. Chabot[1], Sidney Chocron[15], Gareth S. Collins[16], R. Terik Daly[1], Thomas M. Davison[16], Mallory E. DeCoster[1], Carolyn M. Ernst[1], Fabio Ferrari[17], Dawn M. Graninger[1], Seth A. Jacobson[18], Martin Jutzi[5], Kathryn M. Kumamoto[19], Robert Luther[20], Joshua R. Lyzhoft[3], Patrick Michel[21], Naomi Murdoch[22], Ryota Nakano[11], Eric Palmer[12], Andrew S. Rivkin[1], Daniel J. Scheeres[4], Angela M. Stickle[1], Jessica M. Sunshine[2], Josep M. Trigo-Rodriguez[23], Jean-Baptiste Vincent[24], James D. Walker[14], Kai Wünnemann[20,25], Yun Zhang[26], Marilena Amoroso[8], Ivano Bertini[27,28], John R. Brucato[29], Andrea Capannolo[30], Gabriele Cremonese[31], Massimo Dall'Ora[32], Prasanna J. D. Deshapriya[6], Igor Gai[33], Pedro H. Hasselmann[6], Simone Ieva[6], Gabriele Impresario[8], Stavro L. Ivanovski[34], Michèle Lavagna[17], Alice Lucchetti[31], Elena M. Epifani[6], Dario Modenini[33], Maurizio Pajola[31], Pasquale Palumbo[27], Davide Perna[6], Simone Pirrotta[8], Giovanni Poggiali[29], Alessandro Rossi[35], Paolo Tortora[33], Marco Zannoni[33] & Giovanni Zanotti[17]

The NASA Double Asteroid Redirection Test (DART) mission performed a kinetic impact on asteroid Dimorphos, the satellite of the binary asteroid (65803) Didymos, at 23:14 UTC on 26 September 2022 as a planetary defence test[1]. DART was the first hypervelocity impact experiment on an asteroid at size and velocity scales relevant to planetary defence, intended to validate kinetic impact as a means of asteroid deflection. Here we report a determination of the momentum transferred to an asteroid by kinetic impact. On the basis of the change in the binary orbit period[2], we find an instantaneous reduction in Dimorphos's along-track orbital velocity component of $2.70 \pm 0.10$ mm s$^{-1}$, indicating enhanced momentum transfer due to recoil from ejecta streams produced by the impact[3,4]. For a Dimorphos bulk density range of 1,500 to 3,300 kg m$^{-3}$, we find that the expected value of the momentum enhancement factor, $\beta$, ranges between 2.2 and 4.9, depending on the mass of Dimorphos. If Dimorphos and Didymos are assumed to have equal densities of 2,400 kg m$^{-3}$, $\beta = 3.61^{+0.19}_{-0.25}$ ($1\sigma$). These $\beta$ values indicate that substantially more momentum was transferred to Dimorphos from the escaping impact ejecta than was incident with DART. Therefore, the DART kinetic impact was highly effective in deflecting the asteroid Dimorphos.

Observations from the NASA Double Asteroid Redirection Test (DART) spacecraft on approach found Dimorphos to be an oblate spheroid with a boulder-strewn surface, and the spacecraft struck within 25 m of the centre of the figure[1]. Ejecta from the DART impact were observed in situ by the Italian Space Agency's Light Italian Cubesat for Imaging of Asteroids (LICIACube) spacecraft, which performed a flyby of Dimorphos with a closest approach about 168 s after the DART impact[5]. The impact ejecta were further observed by Earth- and space-based telescopes, revealing ejecta streams and dust tails similar to those seen in active asteroids thought to be triggered by natural impacts[3,6]. Ground-based telescopes and radar determined that the DART impact reduced the binary orbit period by $33.0 \pm 1.0$ ($3\sigma$) min (ref. 2).

As a planetary defence test mission, a key objective of DART is to determine the amount of momentum transferred to the target body relative to the incident momentum of the spacecraft, quantified by the momentum enhancement factor $\beta$ (for example, refs. 4,7,8), which is defined by the momentum balance of the kinetic impact,

$$M\Delta\mathbf{v} = m\mathbf{U} + m(\beta - 1)(\hat{\mathbf{E}} \cdot \mathbf{U})\hat{\mathbf{E}}. \quad (1)$$

Here, $M$ is the mass of Dimorphos, $\Delta\mathbf{v}$ is the impact-induced change in Dimorphos's orbital velocity, $m$ is DART's mass at impact, $\mathbf{U}$ is DART's velocity relative to Dimorphos at impact and $\hat{\mathbf{E}}$ is the net ejecta momentum direction. $M\Delta\mathbf{v}$ is the momentum transferred to Dimorphos, $m\mathbf{U}$ is DART's incident momentum and the final term in the equation is the

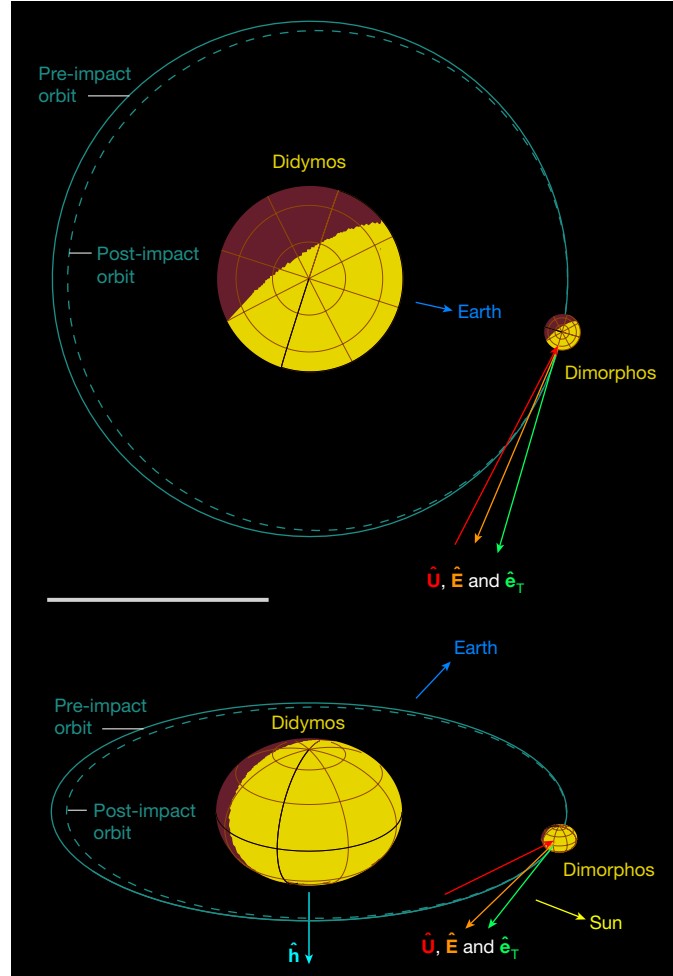

**Fig. 1 | Schematic of the DART impact geometry on Dimorphos.** The pre-impact orbit is shown with a solid line around Didymos. The dashed line sketches the orbit change due to the impact. Orbits are drawn roughly to scale. The positive pole direction of Didymos is $\hat{\mathbf{h}}$ (pointing down in the bottom panel). DART's incident direction is $\hat{\mathbf{U}}$, the net ejecta momentum direction is $\hat{\mathbf{E}}$ (which points to a right ascension (RA) and declination (Dec) of 138° and +13°, respectively), and the direction of Dimorphos's orbital motion, referred to as the along-track direction, is $\hat{\mathbf{e}}_T$. The relative positions of the Sun and the Earth are also indicated. The upper panel shows the view from Didymos's negative pole direction, whereas the lower panel provides a perspective view. Scale bar, 1 km.

$$\beta = 1 + \frac{\frac{M}{m}(\Delta\mathbf{v}\cdot\hat{\mathbf{e}}_T) - (\mathbf{U}\cdot\hat{\mathbf{e}}_T)}{(\hat{\mathbf{E}}\cdot\mathbf{U})(\hat{\mathbf{E}}\cdot\hat{\mathbf{e}}_T)}. \qquad (2)$$

For the remainder of this work, we refer to the along-track component of Dimorphos's velocity change, $\Delta\mathbf{v}\cdot\hat{\mathbf{e}}_T$, as $\Delta v_T$. Figure 1 shows the geometry of the DART impact, including the nominal ellipsoidal shapes used for Didymos and Dimorphos in our analysis, and the nominal orientations of $\mathbf{U}$, $\hat{\mathbf{e}}_T$ and $\hat{\mathbf{E}}$ at the time of impact.

The major unknowns in calculating $\beta$ are $\Delta v_T$, $M$ and $\hat{\mathbf{E}}$. We first use a Monte Carlo approach to produce a distribution for $\Delta v_T$ consistent with the measured period change that incorporates the various uncertainties involved. We sample many possible combinations of Didymos system parameters, including the ellipsoid shape extents of the asteroids, pre-impact orbit separation distance between the two asteroids' centres of mass (that is, Dimorphos's pre-impact orbit radius), pre- and post-impact orbit periods, and net ejecta momentum direction $\hat{\mathbf{E}}$. We use the full two-body problem code (General Use Binary Asteroid Simulator (GUBAS)[11], Methods) that implements coupled rotational and orbital dynamics to numerically determine $\Delta v_T$ for each sampled combination of input parameters. Coupled dynamics are necessary because of the non-spherical shapes of Didymos and Dimorphos and their close proximity relative to their sizes. A range of values for $M$ is generated by combining the volumes of the sampled ellipsoid shape parameters with values for Dimorphos's density. The Monte Carlo approach is summarized by Extended Data Fig. 1. As Dimorphos's density has not been directly measured and has a large uncertainty, we treat it as an independent variable and uniformly sample a wide range of possible values between 1,500 and 3,300 kg m$^{-3}$, a range that encompasses the 3$\sigma$ uncertainty[1]. Using a technique modified from that of ref. 12 (Methods), we apply observations of the ejecta by means of Hubble and LICIACube data to obtain a preliminary measurement of the axis of the ejecta cone geometry (Extended Data Figs. 2 and 3). The cone axis direction is identical to $\hat{\mathbf{E}}$ assuming the ejecta plume holds the momentum uniformly, and we find $\hat{\mathbf{E}}$ points towards a right ascension and declination (Dec) of 138° and +13°, respectively (Extended Data Fig. 2). We assign a conservative uncertainty of 15° around this direction. Finally, $\beta$ also depends on DART's mass and impact velocity, as well as Didymos's pole orientation[2]. Those quantities have negligibly small uncertainties relative to those of the other parameters discussed previously and are therefore treated as fixed values (not sampled). See Methods for additional details on the Monte Carlo analysis, Extended Data Table 1 for a list of parameters and uncertainties, and Extended Data Table 2 for the covariances that were used.

We find that $\Delta v_T = -2.70 \pm 0.10$ (1$\sigma$) mm s$^{-1}$, on the basis of the observed impact-induced period change of $-33.0 \pm 1.0$ (3$\sigma$) minutes and the shapes and separation of Didymos and Dimorphos[1,2]. Figure 2 shows the distribution of $\Delta v_T$ values from the Monte Carlo analysis, along with the fitted mean and standard deviation. The resulting spread of $\beta$ values as a function of Dimorphos's density, calculated by means of equation (2), is presented in Fig. 3, along with linear fits for the mean $\beta$ versus density trend and its 1$\sigma$ confidence intervals. The linear-fit slope is expressed as a scale factor on the ratio of density to the nominal value of 2,400 kg m$^{-3}$ (ref. 1). For that nominal Dimorphos density, at which Dimorphos and Didymos would have roughly equal densities, $\beta = 3.61^{+0.19}_{-0.25}$ with 1$\sigma$ confidence. The mean $\beta$ ranges between 2.2 and 4.9 as a function of density across the range of 1,500 to 3,300 kg m$^{-3}$ and, overall, $\beta$ ranges between 1.9 and 5.5 with 3$\sigma$ confidence.

Our result for $\beta$ is consistent with numerical simulations[13–23] and laboratory experiments[24–29] of kinetic impacts, which have consistently indicated that $\beta$ is expected to fall between about 1 and 6. However, non-unique combinations of asteroid mechanical properties (for example, cohesive strength, porosity and friction angle) can

ejecta's net momentum written in terms of the spacecraft incident momentum. In this formulation, $\beta$ is the ratio of actual imparted momentum to the impactor's momentum in the direction of the net ejecta momentum. Although previous works have defined $\beta$ using the impactor's momentum in the surface normal direction[8,9], we elect to use the ejecta direction instead as our reference for the result to be independent of the surface topography. These definitions are equivalent in the case in which the ejecta direction is in the surface normal direction. A $\beta$ value near 1 would indicate that ejecta recoil had made only a negligible contribution to the momentum transfer. A $\beta > 2$ would mean that the ejecta momentum contribution exceeded the incident momentum from DART.

The full $\Delta\mathbf{v}$ cannot be determined with the available information[10], but its component along Dimorphos's orbital velocity direction, referred to as the along-track direction, can be estimated from available data including Dimorphos's orbit period change. To express $\beta$ in terms of the along-track component of $\Delta\mathbf{v}$, we take the scalar product of (1) with the unit vector $\hat{\mathbf{e}}_T$ in the along-track direction. Solving for $\beta$ yields,

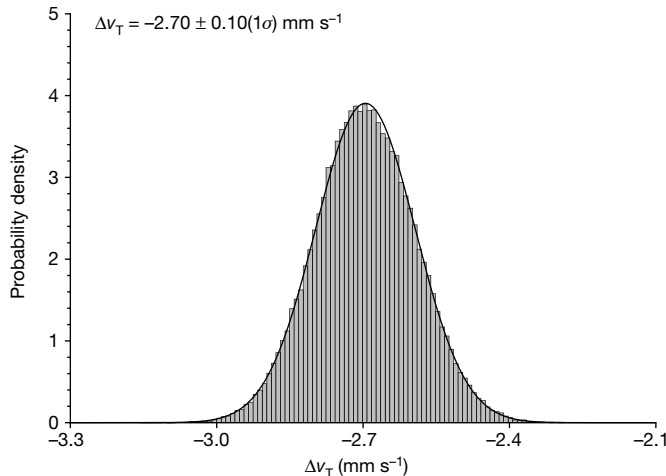

$$\Delta v_T = -2.70 \pm 0.10\,(1\sigma)\ \mathrm{mm\ s^{-1}}$$

**Fig. 2 | Probability distribution of $\Delta v_T$, the along-track component of the change in Dimorphos's velocity induced by DART's impact, generated by our Monte Carlo analysis that samples over input parameter uncertainties.** The histogram consists of 100,000 Monte Carlo samples and is normalized to an area of unity. A Gaussian fit to the distribution indicates a mean $\Delta v_T$ of $-2.70\ \mathrm{mm\ s^{-1}}$ with a standard deviation of $0.10\ \mathrm{mm\ s^{-1}}$.

produce similar values of $\beta$ in impact simulations[20]. Future studies that combine estimates of $\beta$ with additional constraints from the DART impact site geology[1] and ejecta observations[3,5] will provide greater insight into Dimorphos's material properties. In addition, ESA's Hera mission[30] is planned to arrive at the Didymos system in late 2026. By measuring Dimorphos's mass and other orbital properties, Hera will allow us to significantly improve the accuracy and precision of the $\beta$ determination.

DART's impact demonstrates that the momentum transfer to a target asteroid can substantially exceed the incident momentum of the

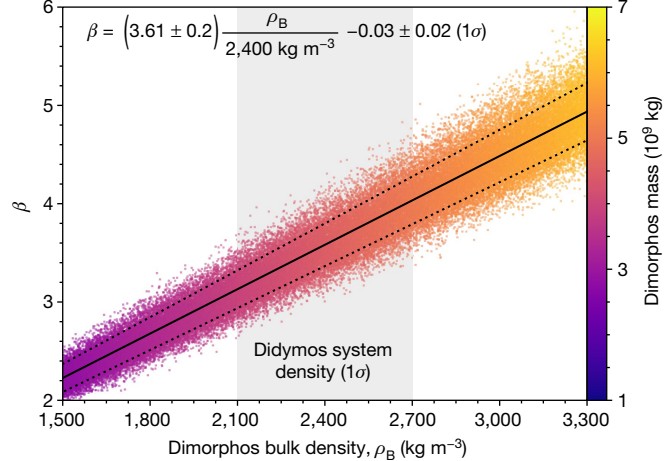

$$\beta = \left(3.61 \pm 0.2\right)\frac{\rho_B}{2{,}400\ \mathrm{kg\ m^{-3}}} - 0.03 \pm 0.02\ (1\sigma)$$

**Fig. 3 | $\beta$ as a function of Dimorphos's bulk density $\rho_B$, from the dynamical Monte Carlo analysis.** Individual samples are plotted as points, whereas the linear fit for the mean $\beta$ is plotted as the solid line and the dotted lines show the $1\sigma$ confidence interval. The colour bar indicates the mass of Dimorphos corresponding to each Monte Carlo sample, which is determined by bulk density and the volume. The density range shown corresponds to the $3\sigma$ range of the Didymos system density, whereas the shaded region highlights the $1\sigma$ range[1]. If the density of Dimorphos were $2{,}400\ \mathrm{kg\ m^{-3}}$, the densities of Didymos and Dimorphos would be the same as the system density, and $\beta = 3.61^{+0.19}_{-0.25}\ (1\sigma)$. For context, the densities of three other S-type near-Earth asteroids are in the range shown: 433 Eros[32] at $2{,}670 \pm 30\ \mathrm{kg\ m^{-3}}$; 25143 Itokawa[33] at $1{,}900 \pm 130\ \mathrm{kg\ m^{-3}}$ and 66391 Moshup[34] at $1{,}970 \pm 240\ \mathrm{kg\ m^{-3}}$.

kinetic impactor, validating the effectiveness of kinetic impact for preventing future asteroid strikes on the Earth. The value of $\beta$ from a kinetic impact is key to informing the strategy of a kinetic impactor mission (or missions) to mitigate a future asteroid impact threat to Earth[31]. Should $\beta$ turn out to be greater than two across a wide range of asteroid types, it would mean important performance improvements for kinetic impactor asteroid deflection missions. If $\beta > 2$, as opposed to $\beta \cong 1$, then the same sized kinetic impactor could deflect a given asteroid with less warning time, or deflect a larger asteroid with a given warning time than it otherwise could.

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

[1]Applied Physics Laboratory, Johns Hopkins University, Laurel, MD, USA. [2]Department of Astronomy, University of Maryland, College Park, MD, USA. [3]NASA/Goddard Space Flight Center, Greenbelt, MD, USA. [4]Smead Department of Aerospace Engineering Sciences, University of Colorado Boulder, Boulder, CO, USA. [5]Space Research and Planetary Sciences, Physical Institute, University of Bern, Bern, Switzerland. [6]INAF, Astronomical Observatory of Rome, Rome, Italy. [7]Space Science Data Center (ASI), Roma, Italy. [8]Italian Space Agency – ASI, Sede di Roma, Rome, Italy. [9]INAF, Institute of Space Astrophysics and Planetology, Roma, Italy. [10]Planetary Defense Coordination Office and Planetary Science Division, NASA Headquarters, Washington, DC, USA. [11]Jet Propulsion Laboratory, California Institute of Technology, Pasadena, CA, USA. [12]Auburn University, Auburn, AL, USA. [13]Planetary Science Institute, Tucson, AZ, USA. [14]Department of Aerospace Engineering, University of Illinois at Urbana-Champaign, Champaign, IL, USA. [15]Southwest Research Institute, San Antonio, TX, USA. [16]Imperial College London, London, UK. [17]Department of Aerospace Science and Technology, Polytechnic University of Milan, Milano, Italy. [18]Michigan State University, East Lansing, MI, USA. [19]Lawrence Livermore National Laboratory, Livermore, CA, USA. [20]Natural History Museum, Leibniz Institute for Evolution and Biodiversity Science, Berlin, Germany. [21]Observatory of the Côte d'Azur, CNRS, Lagrange Laboratory, University of the Côte d'Azur, Nice, France. [22]Higher Institute of Aeronautics and Space (ISAE-SUPAERO), University of Toulouse, Toulouse, France. [23]Institute of Space Sciences (CSIC-IEEC), Barcelona, Spain. [24]DLR Institute of Planetary Research, Berlin, Germany. [25]Freie University of Berlin, Berlin, Germany. [26]Department of Aerospace Engineering, University of Maryland, College Park, MD, USA. [27]Department of Science and Technology, University of Naples 'Parthenope', Naples, Italy. [28]Institute for Space Astrophysics and Planetology (IAPS), INAF, Rome, Italy. [29]INAF, Astrophysical Observatory of Arcetri, Firenze, Italy. [30]Department of Aerospace Science and Technology (DAER), Polytechnic University of Milan, Milan, Italy. [31]INAF, Astronomical Observatory at Padova, Padova, Italy. [32]INAF, Astronomical Observatory at Capodimonte, Napoli, Italy. [33]Department of Industrial Engineering, Alma Mater Studiorum - University of Bologna, Forlì, Italy. [34]INAF, Astronomical Observatory at Trieste, Trieste, Italy. [35]IFAC, CNR, Florence, Italy. ✉e-mail: andy.cheng@jhuapl.edu

## Methods

### Numerical determination of $\Delta v_T$ and $\beta$

Several parameters affect the value of $\beta$ as presented in equation (2): $\Delta v_T$, $M$ and $\hat{\mathbf{E}}$. The along-track velocity change, $\Delta v_T$ depends on orbit period change, pre-impact semimajor axis and the shapes of Didymos and Dimorphos, whereas $M$ depends on Dimorphos's shape and bulk density (which was not measured). $\hat{\mathbf{E}}$ is the only parameter that is directly observed, but it still has considerable uncertainty. Thus, there are 12 total unknown input parameters: three axis lengths for Didymos's ellipsoidal shape ($A_x, A_y, A_z$), three axial lengths for Dimorphos's ellipsoidal shape ($B_x, B_y, B_z$), Dimorphos's bulk density $\rho_B$, the pre-impact orbit semimajor axis $a_{pre}$, pre-impact orbit period $P_{pre}$ and post-impact orbit period $P_{post}$, and two angles to define the ejecta momentum direction vector ($\hat{\mathbf{E}}$). Extended Data Table 1 lists these input parameter values and their uncertainties, along with additional known quantities needed to calculate $\beta$. To account for this large set of input uncertainties, we use a Monte Carlo approach in which 100,000 possible cases are generated by randomly sampling the input parameters within their uncertainties. We assume the DART spacecraft mass, DART impact velocity vector and Dimorphos's orbital velocity direction (referred to as the along-track direction) are all known precisely because their uncertainties are negligibly small compared to the uncertainties of the other input parameters.

The pre-impact orbit semimajor axis, pre-impact orbit period and post-impact orbit period are sampled as a multivariate Gaussian distribution using the mean values and covariance matrix from the 'N22+' solution (refs. 35 and 36 of ref. 2; Extended Data Tables 1 and 2). This accounts for the small correlations between those three parameters. The physical extents of Didymos and Dimorphos from ref. 1 are sampled uniformly, as those uncertainties are not Gaussian (Extended Data Table 1). $\beta$ depends strongly on Dimorphos's mass, but the mass is poorly constrained because Dimorphos's bulk density has not been directly measured[1]. Therefore, we treat density as the independent variable, sample it uniformly and report $\beta$ as a function of Dimorphos's density.

For each Monte Carlo sample for the Didymos system, a secant search algorithm (a finite-difference Newton's method) described in ref. 37 is first used to compute the density of Didymos required to reproduce the sampled pre-impact orbit period, given the sampled pre-impact orbit semimajor axis, body shapes and Dimorphos's density. Then, a second secant search algorithm is used to determine the $\Delta v_T$ required to achieve the sampled post-impact orbit period. We match the pre- and post-impact orbit periods because these are directly measured by ground-based observations and are thus the best-constrained parameters of the system[2]. Given the non-Keplerian nature of the Didymos system, we use the GUBAS to numerically propagate the binary asteroid dynamics. GUBAS is a well-tested full two-body problem (F2BP) code that can model the mutual gravitational interactions between two arbitrarily shaped rigid bodies with uniform mass distributions[11,38]. GUBAS has been benchmarked against other F2BP codes[39] and used extensively in previous dynamical studies of the Didymos system (for example, refs. 10,37,40). Finally, the mass of Dimorphos is calculated from its ellipsoidal shape and Dimorphos's density. This mass, along with the computed $\Delta v_T$ and sampled net ejecta momentum direction, are provided as inputs to equation (2) to calculate the value of $\beta$ corresponding to each of the 100,000 realizations of the system. For a discussion on estimating $\hat{\mathbf{E}}$, see the Ejecta plume direction section below. The process described herein is summarized graphically in Extended Data Fig. 1.

The convergence criteria on both secant algorithms are set such that the simulated orbit period matches the desired orbit period to an accuracy ten times better than the uncertainty on the measurements themselves. The numerical simulations measure the average orbit period of Dimorphos in an inertial frame over 30 days to account for small fluctuations in the mutual orbit period resulting from spin-orbit coupling[40]. Our selection of 100,000 as the number of samples to use in the Monte Carlo analysis was informed by calculating an estimated minimum necessary number of samples from the Central Limit Theorem and then testing sample sizes near that estimated value. The $\beta$ estimate results are well converged with 100,000 samples.

In the numerical simulations, both Didymos and Dimorphos are modelled as triaxial ellipsoids with physical extents from ref. 1. Images from DRACO and LICIACube showed that both Didymos and Dimorphos have an oblate spheroid shape[1]. There is no advantage to using more sophisticated shape models while the internal mass distributions of the bodies are unknown. Instead, the ellipsoidal approximation allows for easy sampling of a range of plausible moments of inertia as a proxy for different internal density distributions. For example, given the current uncertainties in Didymos's physical extents[1], sampling over the given range of ellipsoidal shapes results in a range of plausible second-order gravity terms (analogous to the spherical harmonic terms $J_2$, $C_{22}$ and so on), which play an important role in the system's dynamics due to the tight separation of the binary components. Neglecting their shapes and assuming Keplerian dynamics results in $\Delta v_T = -2.86 \pm 0.095$ ($1\sigma$), whereas GUBAS's second-order gravity model finds $\Delta v_T = -2.70 \pm 0.10$ ($1\sigma$). Although fourth-order dynamics influence higher-order dynamical effects[37,40], we find that it comes with a significantly higher computational cost yet plays a negligible role in determining $\Delta v_T$. A smaller batch (due to increased computational cost) of roughly 4,000 runs was conducted with fourth-order dynamics, which resulted in $\Delta v_T = -2.68 \pm 0.10$ ($1\sigma$), indicating the second-order dynamics model is appropriate for determining $\Delta v_T$ given the current uncertainties in the orbit solution and body shapes. This result was also independently verified using analytical models, accounting for Didymos's gravitational quadrupole, which agreed within a few percent of the second-order numerical results, as expected given their dynamical approximations.

We do not sample the rotation period of Dimorphos, as it is assumed to be equal to the pre-impact orbit period before the impact, with reasoning as follows. A measured orbit semimajor axis drift directed inwards[41] indicates the system is evolving under the influence of the binary Yarkovsky–O'Keefe–Radzievskii–Paddack effect[42], which requires a secondary in near-synchronous rotation. Furthermore, radar images constrain Dimorphos's spin period to be within 3 h of the synchronous rate[35]. Recent models for tidal dissipation in binary asteroids suggest that any free libration would dissipate on 100-year timescales[43], making any substantial free libration unlikely given the timescales for excitation mechanisms such as close planetary encounters and natural impacts[10]. Furthermore, Dimorphos's pre-impact eccentricity is constrained to be less than 0.03 (refs. 35,41), putting the maximum possible forced libration amplitude[44] at around 0.5°. Although Dimorphos's rotation state is not precisely determined by DART, this body of evidence suggests that Dimorphos was probably in near-synchronous rotation and on a nearly circular orbit before the DART impact.

Our model further assumes all momentum is transferred instantaneously, because earlier work showed the time duration of the momentum transfer has a negligible effect on the resulting dynamics[10]. The instantaneous torque on Dimorphos due to DART's slightly off-centre impact[1] is also neglected as the corresponding change in Dimorphos's rotation state is small compared to that arising from exciting Dimorphos's eccentricity and libration by the impact[10,37,45]. Finally, the effects of reshaping and mass loss due to cratering and ejecta are also neglected, as these effects are expected to be smaller in magnitude than the current roughly 1 min uncertainty on the post-impact orbit period[46] and will remain poorly constrained until the Hera mission characterizes the Didymos system in 2027 (ref. 30). We leave these higher-order effects for future work once the post-impact orbit solution is refined further.

## Ejecta plume direction

We use observations of the ejecta plume to determine the ejecta momentum direction **Ê**. The conical ejecta plume was imaged by the LICIACube LUKE camera[5] and the Hubble Space Telescope (HST)[3]. We apply a technique used to derive cometary spin poles[12] to estimate the orientation of the ejecta cone axis. Although it is possible to have an asymmetric distribution of ejecta momentum (mass and velocity) within the cone, we assume the cone to be axially symmetric. The approach applies the ejecta cone's bright edges (if captured in an image) to compute the apparent direction of the cone axis projected onto the sky, which is assumed to be the middle of the edges.

For a LICIACube observation, the projected cone axis defines the LICIACube-axis plane in inertial space that contains the line-of-sight and the projected axis. The cone axis can lie anywhere in this plane. The analogous plane HST axis is defined from early HST images of the plume (those taken within 2 h after the impact) that show similar radial velocity to the ejecta in the LICIACube images, indicating it is probably the same ejecta material observed on a larger spatial scale. The intersection of these planes defines the cone axis orientation in three dimensions, but unfortunately the LICIACube- and HST-axis planes are nearly parallel. Thus, these observations do not provide a unique solution but they constrain the axis orientation to a narrow swath of the sky (Extended Data Fig. 2). However, LICIACube LUKE images resolved the ejecta cone and the cone morphology over a large range of viewing angles during the flyby, further constraining the cone axis orientation[3]. During the approach to Dimorphos, the cone was pointed towards LICIACube, with the ejecta obscuring parts of Dimorphos. During recession from Dimorphos, the cone pointed away from Dimorphos and revealed it in silhouette (Extended Data Fig. 3). The tightest constraint on cone orientation would have come from closest-approach images, with the cone axis in the plane-of-sky, as the cone transitioned from pointing towards the observer to pointing away. Unfortunately, Dimorphos and the ejecta cone were outside the LUKE field of view for 13 s around closest approach, and we lack images from the transition.

The resolved LICIACube images are used to eliminate portions of the swath in Extended Data Fig. 2 where the observed cone morphology is inconsistent with those axis orientations. For example, half of the swath is rejected because the axis would be pointed in the opposite direction of what was observed. We also exclude orientations in which the cone would point too close to the line-of-sight during the approach or recession of LICIACube. We find the axis orientation to be (right ascension, Dec) = (138°, +13°). We assign conservative uncertainties of roughly 15° in all directions based on the angular extents of region 5 in Extended Data Fig. 2.

## Data availability

The dynamical simulations were carried out using GUBAS, which is publicly available on Github (https://github.com/meyeralexj/gubas). The Dimorphos orbital velocity direction vector components presented in Extended Data Table 1 were computed using the dimorphos_s501. bsp and sb-65803-198.bsp (Didymos) data files available at https://dart. jhuapl.edu/SPICE_kernels/spk. The DART incident velocity vector components presented in Extended Data Table 1 were computed using those two files in combination with the DART_2022_269_1241_ops_v01_impact. bsp data file, available at the same URL. Data availability at that URL is planned until summer 2023, after which those data may be found at https://naif.jpl.nasa.gov/naif/data.html.

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

**Acknowledgements** We thank S. Marchi, K. T. Ramesh, P. Sanchez, S. Schwartz, J. Steckloff, M. B. Syal and G. Tancredi for their comments on the manuscript. M.Z., I.G., D.M. and P.T. acknowledge D. Lubey, M. Smith and D. Mages for the useful discussions and suggestions regarding the operational navigation of LICIACube. This work was supported by the DART mission, NASA contract no. 80MSFC20D0004. S.D.R. and M.J. acknowledge support from the Swiss National Science Foundation (project number 200021_207359). S.D.R., M.J., R.L., P.M., N.M. and K.W., acknowledge the funding from the European Union's Horizon 2020 research and innovation programme, grant agreement no. 870377 (project no. NEO-MAPP). P.M. acknowledges support from CNES, ESA and the CNRS through the MITI interdisciplinary programmes. N.M. acknowledges support from CNES. E.D., V.D.C., E.M.E., A.R., I.G., P.J.D.D., P.H.H., I.B., A.Z., S.L.I., J.R.B., G.P., A.L., M.P., G.Z., M.A., A.C., G.C., M.D.O., S.I., G.I., M.L., D.M., P.P., D.P., S.P., P.T. and M.Z. acknowledge financial support from Agenzia Spaziale Italiana (ASI, contract no. 2019-31-HH.O). S.E. acknowledges support through NASA grant number 80NSSC22K1173. S.C. and J.D.W. acknowledge support from the Southwest Research Institute's internal research programme. G.S.C. and T.M.D. acknowledges support from the UK Science and Technology Facilities Council grant no. ST/S000615/1. F.F. acknowledges funding from the Swiss National Science Foundation (SNSF) Ambizione grant no. 193346. J.-Y.L. acknowledges the support provided by NASA through grant no. HST-GO-16674 from the Space Telescope Science Institute, which is operated by the Association of Universities for Research in Astronomy, Inc., under NASA contract no. NAS 5-26555, and the support from NASA DART Participating Scientist Program, grant no. 80NSSC21K1131. R.N. acknowledges support from NASA/FINESST (grant no. NNH20ZDA001N). J.M.T.-R. acknowledges financial support from project no. PID2021-128062NB-I00 funded by MCIN/AEI (Spain). Part of this research was carried out at the Jet Propulsion Laboratory, California Institute of Technology, under a contract with the National Aeronautics and Space Administration. Lawrence Livermore National Laboratory is operated by Lawrence Livermore National Security, LLC, for the US Department of Energy, National Nuclear Security Administration under contract nos. DE-AC5207NA27344 and LLNL-JRNL-84276. Simulations were performed on the YORP and ASTRA clusters administered by the Centre for Theory and Computation, part of the Department of Astronomy at the University of Maryland. M.Z., I.G., D.M. and P.T. wish to acknowledge Caltech and the Jet Propulsion Laboratory for granting the University of Bologna a license for an executable version of MONTE Project Edition S/W.

**Author contributions** A.F.C. led the overall conception and writing of the study. H.F.A. contributed the GUBAS dynamical simulations for the Monte Carlo analysis. B.W.B. contributed the Monte Carlo analysis. A.J.M. validated the GUBAS dynamical simulations. T.L.F. and M.H. determined the ejecta cone orientation. S.D.R. contributed to discussions of β determination and implications. A.F.C., H.F.A., B.W.B., A.J.M., T.L.F. and D.C.R. wrote most of the manuscript text. E.D. led the LICIACube investigation as Principal Investigator. A.Z. led production of LICIACube imaging data. V.D.C. led the calibration of the LICIACube imagers. T.S.S. helped with verification of the GUBAS results and provided comments on the manuscript. S.C. and S.P.N. provided the Dimorphos orbital velocity vector direction, DART velocity vector at impact and the covariance matrix for the pre-impact orbit semimajor axis, mean motion and change in mean motion. J.-Y.L. helped with efforts to estimate the ejecta momentum direction from HST and LICIACube observations. S.E., O.S.B., N.L.C., S.C., G.S.C., R.T.D., T.M.D., M.E.D., C.M.E., F.F., D.M.G., S.A.J., M.J., K.M.K., J.-Y.L., R.L., P.M., N.M., R.N., E.P., A.S.R., D.J.S., A.M.S., J.M.S., J.M.T.-R., J.-B.V., J.D.W., K.W. and Y.Z. provided useful inputs and/or comments on the manuscript. J.R.L. assisted with verification efforts for the numerical simulations. E.D., V.D.C., E.M.E., A.R., I.G., P.J.D.D., P.H.H., I.B., A.Z., S.L.I., J.R.B., G.P., A.L., M.P., G.Z., M.A., A.C., G.C., M.D.O., S.I., G.I., M.L., D.M., P.P., D.P., S.P., P.T. and M.Z. contributed to the development and operation of LICIACube, in addition to provision of LICIACube imaging data.

**Competing interests** The authors declare no competing interests.

**Additional information**
**Correspondence and requests for materials** should be addressed to Andrew F. Cheng.

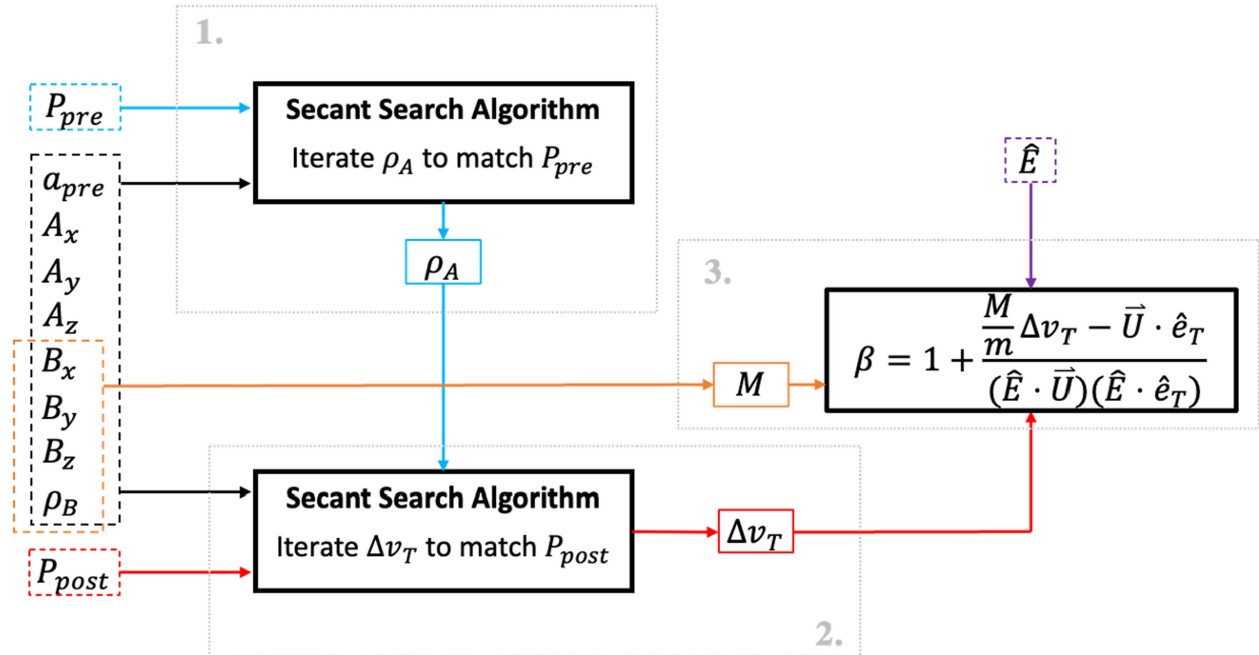

**Extended Data Fig. 1 | Outline of algorithm used to calculate $\beta$.** The Monte Carlo variables are outlined by dashed lines and are defined as follows: pre-impact orbit period $P_{pre}$; pre-impact orbit semimajor axis $a_{pre}$; Didymos ellipsoid extents $A_x, A_y, A_z$; Dimorphos ellipsoid extents $B_x, B_y, B_z$; Dimorphos density $\rho_B$; post-impact orbit period $P_{post}$; and net ejecta momentum direction $\hat{E}$. First, a secant algorithm iterates Didymos density $\rho_A$ to match $P_{pre}$. Next, another secant algorithm iterates the along-track change in Dimorphos's velocity $\Delta v_T$ to match $P_{post}$. Finally, $M$ is calculated using the ellipsoid extents and density of Dimorphos, and then combined with $\Delta v_T$ and $\hat{E}$ to calculate $\beta$.

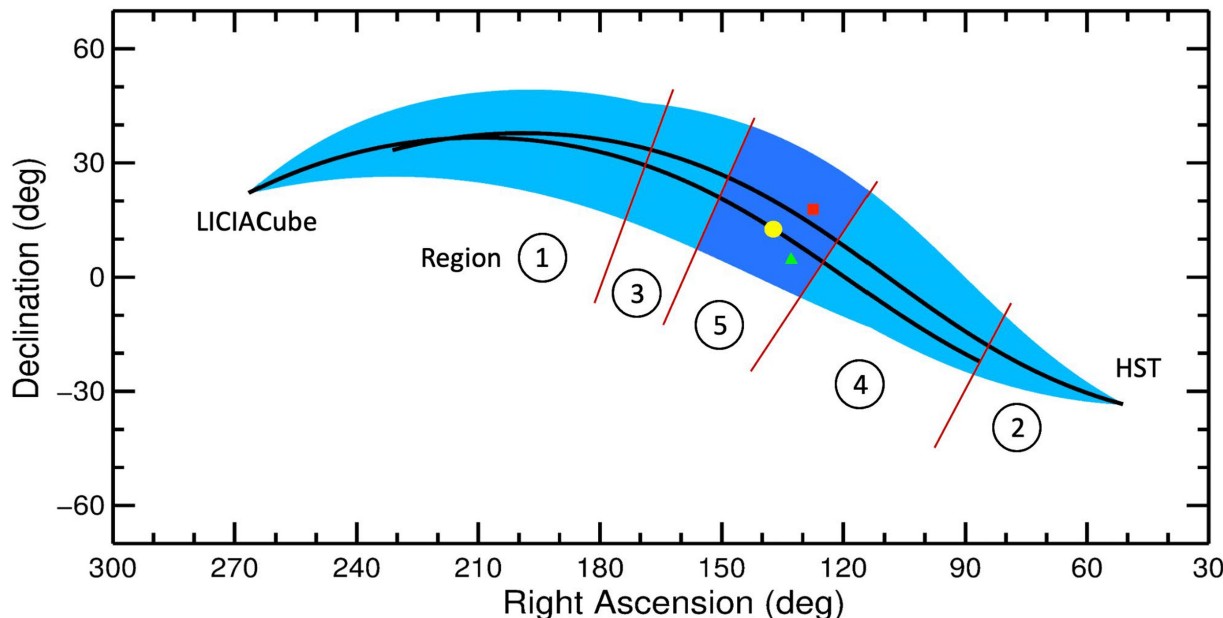

**Extended Data Fig. 2 | Ejecta cone orientation lies in the swaths of sky (black lines) defined by HST and LICIACube observations.** The light-blue envelope outlines the axis position uncertainty in the direction measured in the sky plane. Red lines divide the along-plane swaths into regions that are excluded based on cone morphology in LICIACube images: 1) and 2) are excluded because the ejecta cone would point in the opposite direction from what is observed; 3) is excluded because the axis would lie too close to the sky plane; 4) is excluded because the axis would lie too close to the line-of-sight; and 5) is the expected region for the axis orientation. The yellow dot denotes the best solution (RA,Dec) = [138°,+13°] with the dark-blue envelope showing the extent of possible solutions. The red square is the direction of the incoming DART trajectory [128°,+18°] and the green triangle shows Dimorphos's velocity vector [134°,+5°]. The LICIACube swath is defined for the +175 s image shown in Extended Data Fig. 3.

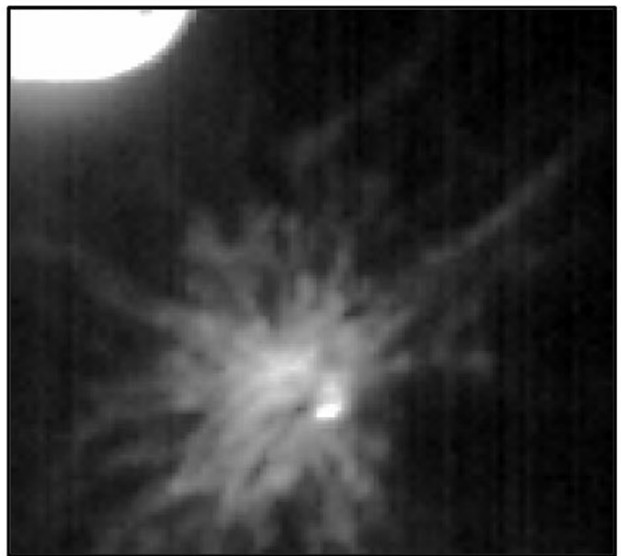 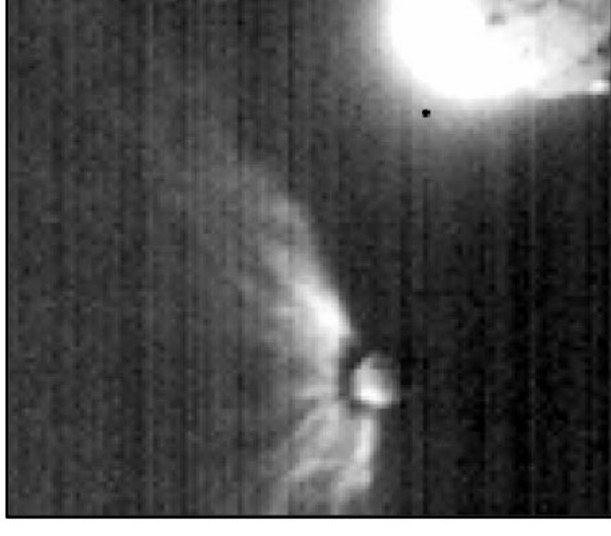

**Extended Data Fig. 3 | Two LICIACube LUKE images showing the ejecta morphology that were used to reduce the possible axis orientation solutions.** The left panel shows an approach observation, 156 s after impact, with the ejecta in front of and partially obscuring Dimorphos. The right panel shows the ejecta morphology after close approach, 175 s after impact, with Dimorphos silhouetted against the ejecta cone. The images show the red channel from frames LICIACUBE_LUKE_L2_1664234220_00005_01 and LICIACUBE_LUKE_L2_1664234239_01003_01. The bright object in the upper corner of each image is Didymos.

**Extended Data Table 1 | Values and uncertainties used for numerical simulations**

| Quantity | Value | Uncertainty Assumed in Monte Carlo simulations | Note |
|---|---|---|---|
| DART mass at impact, $m$ [kg] | 579.4 | Not considered | Actual uncertainty[1] $\pm0.7$ |
| DART incident velocity vector x [km/s], $\vec{U_x}$ | 3.57399 | Not considered | See Data Availability for source information. |
| DART incident velocity vector y [km/s], $\vec{U_y}$ | -4.64106 | Not considered | See Data Availability for source information. |
| DART incident velocity vector z [km/s], $\vec{U_z}$ | -1.85622 | Not considered | See Data Availability for source information. |
| Orbit period pre-impact, $P_{pre}$ [hr] | 11.92148 | $\pm0.000044$ | 1σ, Gaussian, reported by the "N22+" solution in ref. [2] |
| Orbit period post-impact, $P_{post}$ [hr] | 11.372 | $\pm0.0055$ | 1σ, Gaussian, reported by the "N22+" solution in ref. [2] |
| Pre-impact orbit separation (semi-major axis), $a_{pre}$ [km] | 1.206 | $\pm0.035$ | 1σ, Gaussian, reported by the "N22+" solution in ref. [2] |
| Didymos major axis, $A_x$ [m] | 851 | $\pm15$ | sampled uniformly[1] |
| Didymos intermediate axis, $A_y$ [m] | 849 | $\pm15$ | sampled uniformly[1] |
| Didymos minor axis, $A_z$ [m] | 620 | $\pm15$ | sampled uniformly[1] |
| Dimorphos major axis, $B_x$ [m] | 177 | $\pm2$ | sampled uniformly[1] |
| Dimorphos intermediate axis, $B_y$ [m] | 174 | $\pm4$ | sampled uniformly[1] |
| Dimorphos minor axis, $B_z$ [m] | 116 | $\pm2$ | sampled uniformly[1] |
| Dimorphos density, $\rho_B$ [kg m$^{-3}$] | 2400 | $\pm900$ | Sampled uniformly. Three times the density uncertainty on the combined Didymos system from ref. [1] |
| Ejecta cone direction x, $\hat{E}_x$ | -0.72410 | Uncertainty is within a 15° cone around this nominal vector | Sample uniformly throughout a cone of directions up to 15° away nominal RA,Dec = 138°,+13° |
| Ejecta cone direction y, $\hat{E}_y$ | 0.65198 | | |
| Ejecta cone direction z, $\hat{E}_z$ | 0.22495 | | |
| Dimorphos orbital velocity direction x, $\hat{e}_{Tx}$ | -0.689282 | Not considered | See Data Availability for source information. |
| Dimorphos orbital velocity direction y, $\hat{e}_{Ty}$ | 0.716645 | Not considered | See Data Availability for source information. |
| Dimorphos orbital velocity direction z, $\hat{e}_{Tz}$ | 0.106350 | Not considered | See Data Availability for source information. |

All vectors are reported in the Earth Mean Equator J2000 (EME J2000) coordinate frame, at the impact time of September 26, 2022, 23:14:24.183 UTC[1]. For Gaussian uncertainties we report the 1σ uncertainties. For uniform uncertainties we report the median and the range of possible values.

**Extended Data Table 2 | Covariance matrix**

|  | $a_{pre}$ | $n_{pre}$ | $\Delta n$ |
|---|---|---|---|
| $a_{pre}$ | 1.23278161e-03 | -5.07112880e-12 | 7.08461748e-10 |
| $n_{pre}$ | -5.07112880e-12 | 2.90452780e-19 | -2.31470184e-17 |
| $\Delta n$ | 7.08461748e-10 | -2.31470184e-17 | 4.94295884e-15 |

The covariances used to sample semimajor axis, $a_{pre}$, and pre- and post-impact orbit periods. The orbital solution from refs. 2,36 fits the pre-impact mean motion at the impact epoch, $n_{pre}$, and the change in mean motion due to the DART impact, $\Delta n$. Once these parameters are sampled, they are turned into a pre- and post-impact orbit period by the relation $P_{pre} = 2\pi/n_{pre}$ and $P_{post} = 2\pi/(n_{pre} + \Delta n)$. The covariance matrix is constructed using units of km for $a_{pre}$ and rad/s for $n_{pre}$ and $\Delta n$.