## [Peer Review File · Nature]

Manuscript Title: Momentum Transfer from the DART Mission Kinetic Impact on Asteroid Dimorphos

Reviewer Comments & Author Rebuttals

Reviewer Reports on the Initial Version:

Referees' comments:

Referee #1 (Remarks to the Author):

A. Summary of key results: This paper uses the measurement of the ~30 minute period decrease for Dimorphos that followed the DART impact to model the momentum enhancement factor relative to a purely inelastic collision. The modeling discussed in this brief article includes the measurement of the ejecta tail's direction vector, and is presented using the relatively unconstrained density of Dimorphos as an independent variable. For likely densities, the momentum enhancement factor is measured to be of order 3.6. This is a key, new, direct experimental result that is of signal importance for the feasibility of Earth-threatening asteroid detection. In short, ejecta enhances the impactor's orbit modifying power, meaning that future deflection missions can be effective at lower impactor mass, or alternately, can be carried out within narrower windows.

B. Given that Dimorphos is the first natural Solar System object to have had its trajectory altered in a measurable way, and given the ramifications of this alteration to planetary defense, this article is of self-evident originality and significance, and would pair well with a study that discusses the details of how the period modification was measured.

C. The Newtonian physics of the DART impact are well-understood, and numerical modeling of the binary asteroid is well-established. The data, in particular the period measurement and the ejecta angle are well characterized, and the discussion of procedure (both in the paper and in the supplemental information) is clear and easy to follow.

D. Given the decision to treat the density as an independent variable, I think that the use of statistics and the treatment of uncertainties is correct and proper. The variation in beta produced by different assumed values for the Dimorphos density is clearly delineated in the bolded first paragraph.

E. The conclusions appear robust, and it is clearly indicated that the residual unknowns will be addressed in detail by the forthcoming ESA Hera Mission.

F. In Figure 1, it is not clear whether the modified orbit is schematic or whether it represents the actual best-effort trajectory of the post-impact Dimorphos. I think that an indication of the physical scale of the system on the plot, as well as an estimate of the eccentricity of the modified orbit would be useful.

G. The references appear appropriate.

H. The paper is clear and easy to read.

Referee #3 (Remarks to the Author):

A. Summary of the key results.

The submitted manuscript reports about the outcome of the DART (NASA) spacecraft impact experiment and derives values for the linear momentum enhancement factor (β).

B. Originality and significance: if not novel, please include reference.

No question about originality of the results, this is the first time in human history that an impact experiment on an asteroid is performed. The outcome is significant for asteroid deflection strategy.

C. Data & methodology: validity of approach, quality of data, quality of presentation.

The methodology used to derive estimation of β is fully consistent and was already described in a former paper (Riven et al, 2020 PSJ). Quality of data is very acceptable, given the experimental and observational conditions.

D. Appropriate use of statistics and treatment of uncertainties.

Correct.

E. Conclusions: robustness, validity, reliability.

The authors are fully aware of the limited accuracy of some of the key parameters, and they draw their conclusions accordingly, providing corresponding uncertainty for the derived values of the linear momentum enhancement factor.

F. Suggested improvements: experiments, data for possible revision.

None.

G. References: appropriate credit to previous work?

References are generally fairly indicated.

Nevertheless, in page 2, when ref. 4 is cited, also reference to a recently published paper should be made: "Lofting of low speed ejecta produced in the DART experiment and production of a dust cloud", by Tancredi et al., MNRAS 2022. They make clear statement to the mechanism of formation of active asteroids in the context of the DART experiment itself.

H. Clarity and context: lucidity of abstract/summary, appropriateness of abstract, introduction and conclusions.

The manuscript is clear and properly written in all its parts.

ADDITIONAL COMMENTS:

- In page 2, on the third line following the abstract, ASI (Italian Space Agency) should be cited somehow like " ... in situ by the ASI's Light Italian Cubesat for Imaging of Asteroids ...".

- In Methods, page 15, in the middle paragraph, considerations are made about the likelihood of near-synchronous rotation of Dimorphos. I believe that this is not robust as there is no compelling evidence supporting that. On the one hand, the argument made is reasonable, on the other hand, arguments against tidal locking can be made. For instance, due to the high eccentricity of the system orbit, Dimorphos spends 1/3 of it inside the inner asteroid belt, where it can undergo low energy collisions (non disruptive) erasing tidal locking from time to time, potentially even introducing wobbling. Simple probability estimations suggest that a few to a few tens such collisions are possible in the typical NEA 1-km size lifetime.

I suggest that such paragraph is reworded so that tidal locking is presented as an assumption based on reasonable considerations, not as a fact. Therefore, it should be pointed out that near-synchronous rotation of Dimorphos may not be the case.

Finally, I congratulate the authors for the excellent work carried out in the interpretation of the DART experiment outcome, and I recommend this manuscript for publication, once the above suggestions are taken into account.

Author Rebuttals to Initial Comments:

Response to Reviewers

We thank the referees for their reviews and for their helpful and supportive comments. We have revised Fig. 1 and its caption, as suggested by Referee #1, and we have revised the discussion paragraph in Methods as requested by Referee #2. Detailed responses follow to the Referees' comments. We wish to participate in transparent peer review.

Referee #1

In Figure 1, it is not clear whether the modified orbit is schematic or whether it represents the actual best-effort trajectory of the post-impact Dimorphos. I think that an indication of the physical scale of the system on the plot, as well as an estimate of the eccentricity of the modified orbit would be useful.

This figure is a schematic illustrating the DART impact geometry on Dimorphos, and the caption has been revised to clarify this point. A scale bar has been added in the figure per this suggestion. However, providing an estimate of the modified orbit's eccentricity would be outside the scope of this work. As of now, there is no consensus on a measured eccentricity of the new orbit, which is the subject of ongoing observational studies and analyses to determine eccentricity together with apsidal precession, which likewise depends on the primary's effective J_2 gravity term. Furthermore, in the non-Keplerian, coupled systems of binary asteroids, there is a difference between the physical eccentricity (calculated from the maximum and minimum separation distances between the two bodies) and the osculating eccentricity, and any eccentricity results would need to be accompanied by a discussion of this difference. Finally, analysis of the post-impact orbit eccentricity is not necessary for the analysis of the impact-induced momentum change that is the focus of this paper. Considering these points, we feel the issue of the new orbit's eccentricity falls outside the scope of this paper.

Referee #2

References are generally fairly indicated.

Nevertheless, in page 2, when ref. 4 is cited, also reference to a recently published paper should be made: "Lofting of low speed ejecta produced in the DART experiment and production of a dust

cloud", by Tancredi et al., MNRAS 2022. They make clear statement to the mechanism of formation of active asteroids in the context of the DART experiment itself.

This reference has been added as ref. 6, per this suggestion.

In page 2, on the third line following the abstract, ASI (Italian Space Agency) should be cited somehow like " ... in situ by the ASI's Light Italian Cubesat for Imaging of Asteroids ...".

The text has been updated with this wording.

In Methods, page 15, in the middle paragraph, considerations are made about the likelihood of near-synchronous rotation of Dimorphos. I believe that this is not robust as there is no compelling evidence supporting that. On the one hand, the argument made is reasonable, on the other hand, arguments against tidal locking can be made. For instance, due to the high eccentricity of the system orbit, Dimorphos spends 1/3 of it inside the inner asteroid belt, where it can undergo low energy collisions (non disruptive) erasing tidal locking from time to time, potentially even introducing wobbling. Simple probability estimations suggest that a few to a few tens such collisions are possible in the typical NEA 1-km size lifetime. I suggest that such paragraph is reworded so that tidal locking is presented as an assumption based on reasonable considerations, not as a fact. Therefore, it should be pointed out that near-synchronous rotation of Dimorphos may not be the case.

In this paragraph we have removed any reference to a 'tidally locked' system in favor of 'near-synchronous,' which is a less rigid term. We believe the arguments laid out in this paragraph, specifically the detection of the BYORP effect and the radar measurements of the secondary's spin rate, make a strong case for a near-synchronous secondary prior to the impact. To address the point regarding impacts in the inner asteroid belt, please see Richardson et al, 2022, Predictions for the Dynamical States of the Didymos System before and after the Planned DART Impact (reference 10 in the manuscript, already cited in the paragraph in question), which presents an argument that the frequency of excitation by impacts is slower than the expected dissipation rate of close-orbiting, rubble-pile binary asteroids. Nevertheless, in the text we only claim Dimorphos was 'likely' in a near-synchronous rotation and state this as an assumption rather than fact, allowing for the possibility that Dimorphos may not be in near-synchronous rotation. Also, we note that this assumption has no significant bearing on the estimation of β that is the focus of this paper.